# Design and Fabrication of Organ-on-Chips: Promises and Challenges

**DOI:** 10.3390/mi12121443

**Published:** 2021-11-25

**Authors:** Alireza Tajeddin, Nur Mustafaoglu

**Affiliations:** 1Faculty of Engineering and Natural Sciences, Sabanci University, Tuzla 34596, Istanbul, Turkey; tajeddin@sabanciuniv.edu; 2Nanotechnology Research and Application Center (SUNUM), Sabanci University, Tuzla 34596, Istanbul, Turkey

**Keywords:** organ-on-chips, microfabrication, microfluidics, 3D organ models

## Abstract

The advent of the miniaturization approach has influenced the research trends in almost all disciplines. Bioengineering is one of the fields benefiting from the new possibilities of microfabrication techniques, especially in cell and tissue culture, disease modeling, and drug discovery. The limitations of existing 2D cell culture techniques, the high time and cost requirements, and the considerable failure rates have led to the idea of 3D cell culture environments capable of providing physiologically relevant tissue functions in vitro. Organ-on-chips are microfluidic devices used in this context as a potential alternative to in vivo animal testing to reduce the cost and time required for drug evaluation. This emerging technology contributes significantly to the development of various research areas, including, but not limited to, tissue engineering and drug discovery. However, it also brings many challenges. Further development of the technology requires interdisciplinary studies as some problems are associated with the materials and their manufacturing techniques. Therefore, in this paper, organ-on-chip technologies are presented, focusing on the design and fabrication requirements. Then, state-of-the-art materials and microfabrication techniques are described in detail to show their advantages and also their limitations. A comparison and identification of gaps for current use and further studies are therefore the subject of the final discussion.

## 1. Introduction

Tracing the history of in vitro research back to its beginnings in the early twentieth century, it was started by Ross G. Harrison when he worked on living nerve fibers [1]. Thereafter, numerous valuable attempts were made to develop techniques to grow cells and observe their differentiation outside of a living organism [2]. Since the 1930s and the discovery of collagen as a component of connective tissue, a number of experiments have made scientific breakthroughs by creating environments for cell growth. This has been accompanied by other interdisciplinary research, including biomechanics and mechatronics, to develop the technology [3,4]. The latest approaches are divided into 2D and 3D culture techniques (Figure 1). In 2D cultures, cells are grown as monolayers on flat surfaces, which are usually glass/polystyrene petri dishes [5]. As there is no mechanical support for the cells other than the flat plate, the shape is not controlled. To improve the shape factors that affect biofunction in vivo, microstructured substrates, such as cell adhesive islands and micro pillars, have been introduced [6]. This shape enhancement helps improve cell function in vitro, which has led researchers to incorporate more advanced environments into cell culture systems to create 3D models and develop techniques, such as sandwich culturing, microstructuring, and substrate stiffness modification [7]. Unlike 2D culture environments, 3D culture allows cells to grow and interact in all dimensions. Scaffold and free-scaffold approaches to control shape are presented [8]. Although 2D cultures allow low-cost assays, they cannot fully recapitulate the overall structure and physiological functions. Several reasons have been suggested for this failure, notably the uncontrolled access to oxygen and other ingredients in monolayer cultures and the lack of cell–cell and cell–matrix interactions [9]. There has been a significant incentive to move from semi-controlled flat environments to controlled 3D forms to fully ensure in vivo physiological properties. 

The “spheroid culture” (Spheroid culture: Multicellular, spherical structures composed of aggregated cells that do not adhere to a substrate but adhere to each other [10]) system is one of the first 3D methods used for studying tumor models, biology, tissue engineering, and transplantation therapies [11,12,13]. There are several techniques for this culturing system, such as hanging drops, pellet culture, and magnetic levitation [14]. Despite the good agreement in capturing cell interactions and the physiochemical environment, there are still some drawbacks related to the dependence on the size of the spheroids [15]. 

Further attempts to find alternative 3D culturing methods that better represent in vivo physiology led to “organoids” (Organoids: A self-organizing 3D cell structure that represents an organ with in vivo-like functions and physiology [16]). To allow easy comparison with spheroids, organoids produce more complex tissues and facilitate the study of tissue-specific functions. They are widely used for various applications, such as disease pathology, drug toxicity, and personalized treatment. On the other hand, the variability and the lack of a uniform protocol for the use of organoids remain one of the challenges of this approach [17,18]. In addition, organoids are not suitable for the study of stromal, vascular, neural, and immune cells because it is not possible to apply mechanobiological stimuli, such as the flow and the “shear stress” (Shear stress: The ratio of the tangential force to the surface area in a channel which is related to the velocity gradient and the viscosity of the fluid [19]), as well as the cyclic strain [20]. Furthermore, it is impossible to model and study the vasculature and apply various shear stresses [21].

With the advent and development of tissue engineering, another approach became interesting, namely the use of “printed tissues” (Printed tissues: Accurate 3D printed tissues/organs through controlled localization of cells and materials [22]). The layer-by-layer deposition of 3D scaffolds with high accuracy at the microscale improves cell response and provides a multifunctional environment for highly efficient cell culturing [23]. However, there are some limitations, such as the lack of precise cell placement and the inability to grow cells at high cell density. Although the production of a vascular network using 3D printing techniques is challenging, emerging technologies and improvements in research have been used to overcome these limitations [24].

Looking at the cell-culturing technologies presented and assessing the arguments for and against, it is clear that there is a need for a more thorough technology. Microfluidics is considered to be the complementary discipline that can make up for any shortcomings and improve upon existing standards. 

Microfluidics can be defined as the study of fluid-flow phenomena in microscale [25]. Originally introduced by the microelectronic industry, the concept was later integrated into other fields ranging from material science to biology [26,27]. Today, more and more applications can be found in the fields of biology, chemistry, the environment, energy, and biomedicine [28,29]. Therefore, microfluidic devices can be considered as one of the rapidly developing fields of science and technology and are increasingly used in many research areas [30]. The precise functioning of microfluidic devices is a significant driver of the observed development in recent decades, especially for the performance of physiologically relevant biological cultures and the development of functional organ tissues in vitro. Recently, pharmaceutical technologies have become proponents of microfluidic applications as microphysiological systems enable faster drug development and better cost management, provide effective drug selection with low risk, and enable effective drug production in human models [31]. To be precise, “Organ-on-chip” (Organ-on-a-chip: Microfluidic devices consisting of multi channels compatible with cell culturing which resembles the physical and physiological functions of a specific organ [32]) (OOC) technology, which generates fully functional in vivo-like organ units in physiologically relevant mechanobiological environments, overcomes the limited resources available for preclinical testing for drug screening and delivery and thus reflects a great inclination towards the use of microfluidic devices [33,34,35,36]. 

OOCs are biocompatible microfluidic devices based on human cells to create human-like test systems [37]. Three-dimensional microfluidic chip models are cultured with cells accompanied by controlled external mechanical parameters to simulate an accurate physiological environment for studying the interactions between cells, tissues, and drugs. This interdisciplinary technology not only recapitulates the basic cellular structure of organs, but also mimics the function of a given organ in vitro [38]. OOCs are widely used to study various organs and tissues, including lung [39,40,41], liver [42,43], intestine [44,45,46], brain [47,48], and blood-brain barrier (BBB) [49,50,51,52], as well as multiple organs together [53,54], enabling many major breakthroughs for the understanding of human cell biology, disease physiology, and drug development, while providing superior alternatives to animal models that often fail to predict clinical trial outcomes [32,55].

Beyond academia and into global health, OOC technologies provide promising results for overcoming the shortcomings of drug development models that are highly dependent on costs and timing. Current preclinical testing mainly uses laboratory animals and faces two main problems; (i) the ethical prohibition and (ii) the different responses to the drugs compared to those of human cells [56]. Many preclinical animal tests lead to failures at the final stage of in vivo testing, resulting in a loss of resources and time [57]. Clearly, microfluidic OOC technologies can be one of the key players in solving this problem, as they can serve as platforms for the precise screening of drugs in in vitro models based on human cells, leading to more effective results in both the treatment and the reduction in side effects [58,59].

Parallel with the growing number of research ideas on integrating organ-on-chips, there are attempts to review the existing literature from diverse perspectives in order to assess, collect, and identify the necessary research goals that are currently lacking in the field. Bo et al. have presented one of the earliest reviews of microfluidic chips that can recapitulate tissue functions [60]. In this review, they discuss how microfluidic systems can be effectively used in cell biology and present some OOC models and their future challenges. Zhang et al. wrote a review about OOC technologies and introduced this technology, with its future prospects, as a solution to clinical translational problems, offering an alternative to traditional preclinical models for drug screening [61]. Eduardo et al. presented an overview of the concepts, fabrication, and recent advances in OOCs [62]. In another study, Wu et al. presented OOCs along with the applications to physiological models, drugs, and toxicology studies for different organs [63]. There are some other reviews that specifically address OOCs for drug evaluation [55], radiology [64], and toxicity [65]. However, there is no comprehensive review that addresses the design, material, and fabrication of OOCs for the perspective of different organs.

In this review, the design and fabrication requirements of OOCs are presented and evaluated individually. First, the design considerations are explained in detail, including the flow control of OOCs, the prevention of “clogging” (Clogging: Interruption of flow due to the aggregation of particles [66]), and the monitoring of the cells inside the chips. The fabrication material and methods were evaluated with their advantages and disadvantages. Different types of available OOCs are then presented to give a general insight into the concept and to assess how the combination of design, material, and fabrication affects their performance. Finally, a thorough discussion was held on the existing knowledge and challenges in order to draw conclusions about the best design and fabrication methods.

## 2. Conceptual Design of OOC

The following provides detailed information on the design considerations for OOC models, including their geometric design and flow control and the design considerations for avoiding clogging in the OOC microfluidic channels, as well as the integration of real-time monitoring systems into the OOC models.

### 2.1. Geometry and Dimensions

There are various chip designs for the study of organs, which differ in size, diameter, number of channels, shape of channels, and other geometrical features. Hence, the design is organ-oriented, and the corresponding features are specified accordingly [67]. Although there is no uniform geometry for all OOC models, OOCs can typically be classified based on the number of their compartments/channels numbers and the organization of these channels: (i) single-channel chips, (ii) double-channel chips, which include parallel designs and sandwich designs, and (iii) multichannel chips, as shown in Figure 2.

Double-channel OOC designs are among the most commonly used designs, in which a centimeter-sized chip OOC consisting of two separate channels connected by a porous membrane or compartmentalization can be achieved by hydrogels to study the interphase between different cells in the same tissue [68]. Often, in the design of the OOCs, one compartment was used to mimic the blood vessels and the other compartment(s) for the actual tissue cells [69]. Blood-brain barrier, intestine, and lung are some of the organs that can be mimicked with this type of chip [49,50,63]. There are two inlets and outlets on the chip to control the entry and exit of the working fluid and also the introduction of the biological materials, such as basal laminal proteins, cells, and therapeutic drugs into the system [70].

Porous membranes are usually polymeric, flat microstructures used to recapitulate the permeability between two environments in order to have cell adhesion and separation as well as make communication in between the two compartments. There are some considerations for membrane properties, including stiffness (to follow morphology), porosity, hydrophilicity, and surface roughness [71]. 

The shapes of channels vary extensively and include circular and rectangular types [72]. The size is commonly in microscale and must be consistent with the research objective [33]. In the study of microvasculature, the size varies from millimeters to submicrometers (Figure 3). The dimensions are also related to the working fluid and clogging problems, which will also be discussed below.

### 2.2. Flow Control in OOCs

The control of flow in OOCs is essential to obtaining accurate results. The flow rate affects stresses (shear), polarity, concentration gradients (oxygen and nutrients), and many other dominant parameters [74]. There are two types of flow in a functioning OOC: (i) steady and (ii) pulsatile. The flow is laminar (low Reynolds values) and the value of the flow rate changes depending on the research objective (2 nL/min–5 mL/min) [75]. For instance, the studies conducted on lung-on-chips show the flow rate of 60 µL/h (vasculature side), which resulted in a shear stress of 0.00017 Pa [76,77]. Table 1 shows the flow rates in different experiments conducted on OOCs. Furthermore, for rectangular microchannels working with laminar Newtonian fluids under steady condition, shear stress can be calculated as per the following equation:(1)$$\tau =\frac{6\mu Q}{wh^{2}}$$
where *µ* is fluid dynamic viscosity (Pa·s), *Q* is the volumetric flow rate (m^3^/s), *w* is the width of the channel (m) which is the surface interacting with the cultured cells, and *h* is the height of the microchannel (m).

Flow in OOCs is controlled by several approaches, including pressure-driven, electroosmotic (electrokinetic flow), surface tension, shear flow, gravity, buoyancy, squeeze film, laser-induced, and biological flow (Table 2). Pressure-controlled flow and electroosmotic flow are the most commonly used in practice [87]. While syringe pumps remain one of the most common devices for controlling flow by pressure, there are other mechanical pumps used to precisely control flow in the microfluidic channels of OOCs, including reciprocating and peristaltic pumps [88].

Microvalves are another important component in maintaining the flow in an OOC. They provide control in multiple unit operation as well as fluid transport, mixing, and separation. Various materials (metals and silicon-based compositions) are used to fabricate them [93]. There are many mechanisms to design a microvalve, including the capillary, check, siphon, and hydrophobic valves, which are categorized as passive approaches [94,95]. Additionally, different types of the active microvalves are applicable, including the pneumatic, piezoelectric, electrostatic, and others [96]. Recent experiments have led to novel methods such as optically controlled valves [97] and the use of digital and automated systems [98].

### 2.3. Clog Avoidance in OOCs

The concurrence of working with micro scales and soft materials presents an obstacle called clogging. Clogging is defined as the interruption of flow due to the aggregation of particles [99]. Although the definition may convey a negative situation, it can be beneficial in some cases, such as in the amplification of porous substrates or in the detection of biological cells [100,101]. However, in general, clogging the flow in OOC culturing systems needs to be solved with various troubleshooting approaches as it usually interrupts the cell growth and negatively effects the functionality of the organ chip model.

In general, there are three mechanisms for clogging in a microchannel: (i) sieving, (ii) bridging, and (iii) aggregation (Figure 4). Sieving occurs when the size of the particles is larger than the dimensions of the channel, although there are exceptions in soft materials such as cells [102]. In contrast to sieving, particles in bridging clogging are smaller than the channel and form an arch-shape along the width of the channel due to the steric effects [103]. Clogging by aggregation results from a continuous deposition of particles. The aggregated layer grows as a result of competition between hydrodynamic, diffusive, and colloidal effects [104,105].

There are experimental and computational approaches to preventing clogging in microfluidic channels for both single-pored and porous materials [106]. Many parameters must be considered when investigating clogging phenomena, including the pressure and forces between the particles themselves and between the particles and the walls of the microchannels [99]. Microfiltration is a common approach to addressing particle clogging, but in the case of the OOCs that require a long operating time, it is not a suitable solution because it leads to filter malfunction after a certain time [107]. To avoid this drawback, which is significant in long-term culture studies, in some cases, a bidirectional micropump is used to wash the filters (porous membranes) to extend the operating time [108]. In addition, in many cases, bubbles are the main reason for clogging; so, the flow dynamics in the microchannels needs to be studied to avoid the bubble formation [109]. Maintaining the right pressure and temperature and controlling roughness, as well as using the bubble traps, are the available methods to reduce and/or remove the bubbles in the organ chip devices [110]. 

### 2.4. Monitoring and Detection

When designing a microfluidic device that mimics the flow and the microphysiological conditions of a desired organ, the packaging and fabrication method must also be considered carefully [111]. Packaging consists of the measurement/detection and protection considerations necessary to conduct the particular research. For example, the validation of a chip after fabrication and the checking for leaks and other malfunctions are conducted through benchmarking data that must be recorded during testing. In addition, a validated chip requires tools to measure and monitor the intended parameters during the experimental process and for the final reporting of the results. Therefore, detection tools are inevitable components of OOCs, which also need to be protected because they are very sensitive. This protection can be achieved if the sensors are part of the chip (integrated) or externally mounted (in the packaging) [112]. 

The parameters being studied in an OOC experiment can be sensed in various ways, including mechanically, thermally, chemically, and magnetically. Sensors are detecting tools which transduce these stimulations to electrical and sometimes optical signals that can be read and measured [113]. Basically, there are two branches of sensors that can be incorporated with the OOC systems: (i) physical sensors for monitoring variables, such as pressure, force, and flow rate and (ii) chemical/biological sensors for measuring variables, such as concentration, pH, and protein interactions [114]. The sensors available in OOCs are employed to monitor culture environment, cell behavior, stimulations, mechanical stimulations, chemical gradients, electrical stimulations, hydrogen-peroxide, glucose, and lactate and a myriad of other types [115,116,117,118]. 

Not all conventional sensing mechanisms are suitable for use in OOCs as many of the low-volume devices are required for online monitoring. This has been explored in the relevant literature, particularly for pH and O_2_ sensors, which are critical for studies of cellular metabolisms [119,120]. An electro-optical sensor system has been introduced as a low-cost module in microfluidic devices [121]. This compact system controls optical transducers and signal acquisition from photodiodes (light intensity). Other approaches also exist, and attempts have been made to improve the sensing systems in OOCs significantly, such as with the measurements of pH and O_2_ values with ruthenium oxide (RuO_x_) electrodes, the outputs of which were promising [122].

Trans-epithelial electrical resistance (“TEER” (TEER: transendothelial/epithelial electrical resistance which demonstrates the permeability of the cellular barriers [123])) measurements are one of the most commonly incorporated sensing systems with OOC devices [124]. TEER impedance sensors are usually transparent electrodes patterned by microfabrication techniques and embedded in chips [125]. They operate by applying a continuous current through transcellular (resistance through the apical and basolateral plasma membrane) and paracellular (resistance through the cell substrate) pathways and analyzing the electrical impedances. TEER values demonstrate the high expression of “tight junction” (Tight Junction: the areas where the membranes of the cells are in contact together and have features such as impermeability to form functional barriers [126]) and adherence junction” (Adherence junction: complex proteins forming a cell–cell and cell–matrix junction that is more basal than tight junctions [127]) proteins of epithelial/endothelial cells [128,129]. 

OOCs also require an accurate record of the assay procedure, which includes biological properties (protein and concentration gradients) and morphological parameters (barriers and interaction) as well as physical parameters (O_2_ and osmolarity), which it is possible to monitor not only by incorporating the sensing systems in the chip devices but also by using fluorescence and confocal microscopy technologies for imaging the chips when transparent chip materials are used [121,130]. Therefore, selecting an optically clear material for designing the OOC devices is a must to improve the monitoring capability of the chips.

## 3. Fabrication Materials

Choosing a suitable material is an important parameter for an efficient OOC. It effects many aspects, including the performance, the monitoring, and even the results of the experiments studied on the chips. In this context, there are some important considerations for choosing materials for the fabrication of the chips, such as optical transparency (imaging capability), gas permeability, non-toxicity to cells, cost, and the manufacturing process (Figure 5). Below are some common materials used in practice for OOC fabrication.

### 3.1. Materials Used in Chip Production

To date, there are just some specific materials that can be used for the structure of OOCs, which are discussed in the following. Mechanical properties and optical transparency are important criteria for this purpose [131].

#### 3.1.1. Polydimethylsiloxane (PDMS)

PDMS is the most common material used for the fabrication of microfluidic devices, and OOCs in particular [132]. It is a silicon-based elastomer and has extremely advantageous properties, namely economic feasibility, transparency, flexibility, oxygen permeability, and biocompatibility. It also shows good compliance with various microfabrication techniques, such as soft lithography or molding [133]. On the other hand, there are some properties that limit the use of PDMS and motivate the search for alternatives. The absorption of hydrophobic molecules is a drawback that negatively affects the results of toxicity, efficacy, and also PK/PD (pharmacokinetics/pharmacodynamics) predictions [134]. It is also fluorescent to some degree (with a refractive index of 1.41 to 1.428 in the wavelength range of 403–633 nm [135]) and unsuitable for working with organic solvents [136]. There are increasing attempts to improve the properties of PDMS-made chips by surface modifications using plasma treatment, UV treatment, and coating [137]. There are various coatings that can reduce the surface energy of PDMS; those include some metals/metal oxides (such as titanium oxide and gold) [138] or sol-gel coating [139] and surface silanization techniques (creation of Si-O-Si bonds by amine, carboxyl, thiol, etc.) [140].

#### 3.1.2. Glass

One of the oldest materials in the development of microfluidic devices is glass. In general, there are three types of glass used in this field: (i) soda lime, (ii) quartz, and (iii) borosilicate. They are a mixture of silicon dioxide (SiO_2_), the base material of glass, with other oxides, such as CaO and MgO [141]. Many advantages have been reported on the use of glass in microfabrication, and OOCs in particular, such as transparency, resistance to mechanical stress, hydrophilicity, and biocompatibility. In addition, glass has been reported to have lower drug absorptivity compared to PDMS [142]. On the other hand, one major problem that can lead to channel plugging is the low gas permeability of glass. Therefore, special care must be taken in the design and fabrication, e.g., by the use of bubble traps/removers [143]. Nevertheless, the low-gas permeability feature of the glass can be advantageous in anaerobic studies [144]. Moreover, many microfabrication techniques are applicable to glass, including photolithography, wet etching, and laser cutting [145]. The main reason for preferring polymeric materials (PDMS) over glass is the high cost of fabrication as, unlike polymers, it cannot be easily molded and requires clean room facilities for each step, which also makes the fabrication processes more time-consuming. In addition, the techniques available (anodic or thermal bonding) for bonding the substrates are more demanding [146]. However, there are certain topics for which the use of glass microfluidics is highly recommended, such as the prediction of PK and PD for drug testing and cell-based assays as PDMS absorbs small hydrophobic molecules [147]. Moreover, for the chips in which the integration of electrodes is needed, glass would generally be a better option as polymers (e.g., PDMS) have low stiffness, or a scaffold is required [148].

#### 3.1.3. Thermoplastics

Recently, thermoplastics have been increasingly proposed for the fabrication of microfluidic devices due to the limitations of PDMS and glass-based chips in terms of surface treatment instability, processing techniques, and the absorption of molecules (PDMS) [149]. There are interesting properties that make thermoplastic polymers attractive for OOCs, including low cost, low density, biocompatibility, and easy fabrication [150]. As they have linear and branched molecules, they are more resistant to pressure and temperature fluctuations, which also makes them chemically stable and suitable for biomedical/biochemical studies [151]. Therefore, microfluidic devices based on polymers such as polymethyl methacrylate (PMMA) or copolymers (COC) have been developed and new microfabrication techniques, such as injection molding, casting, and laser cutting have been introduced [152]. On the other hand, there are some limitations in the use of thermoplastic polymers: (i) not all manufactured polymers are transparent (e.g., polyether ether ketone (PEEK) and polypropylene (PP)), which makes microscopic observation or imaging impossible [153,154]; (ii) some have strong autofluorescence properties and are not suitable for detection purposes [155]; (iii) they have poor gas permeability, which has a negative impact on long-term cell culture (such as OOCs) [156]. For example, Trinh et al. designed and fabricated a lab-on-chip for human cell cultures (Mesenchymal Stem Cells (MSCs)) using PMMA, which has the advantage that it can be bonded quickly with acetic acid followed by UV treatment and low pressure, which is suitable for protecting the metal films (electrodes) in chips [157]. This ease of bonding also makes it preferable to PDMS for integrating hydrogels such as 3D ECM gel to maintain organ permeability [158].

### 3.2. Other Materials Used in OOC Technology

There are several other materials in the literature that are used in the process or within OOCs. We present here only the most important ones that are most widely used in the OOC designs. 

#### 3.2.1. Hydrogels

Hydrogels are another new material in the field of OOCs [159]. They are mainly used as bio-scaffolds for cell culturing which is close to the extracellular matrix (ECM) [160]. In 2011, Sung et al. used the first 3D hydrogel scaffold to study the gastrointestinal tract [161]. These are compact hydrophilic polymer chains that are constructed like a network and connected with a high volume of water to form a gel [162]. They were originally used for tissue engineering; they were then applied in other fields such as the food industry and pharmaceutical biosensors [160,163]. These 3D polymers are able to absorb a high level of water while being insoluble. Basically, there are two types of hydrogels: natural (e.g., gelatin, silk, collagen) and synthetic (e.g., PVA (polyvinyl alcohol), PEG (polyethylene glycol)) [164]. Natural hydrogels have advantageous properties, such as biocompatibility, biodegradability, and low cytotoxicity, but they lack controllable mechanical properties, which is why they are often combined with synthetic hydrogels [165]. Recently, hydrogels have become more attractive because of their high permeability and biocompatibility. In addition, the mechanical properties are very similar to some tissue components (such as the extracellular matrix) and are suitable for long-term studies because they protect biological entities. Various microfabrication techniques can be chosen for hydrogel-based devices, including lithography, 3D printing, and molding [159]. Despite their attractive properties, hydrogels are not widely used in microfluidic devices and especially OOCs because their low stiffness is a major problem in microfabrication processes and also limits their long-term use in research [166]. Nevertheless, there are some examples in the literature of the use of hydrogels in OOCs. Sung et al. used collagen to model a 3D intestine with the real size and density of human intestinal villi [161]. They combined laser ablation and molding techniques (sacrificial methods) to produce microstructure collagen. They claim that their method can be further developed for other in vitro organ models to recapitulate realistic geometries. Skin is another organ that researchers are interested in modeling with hydrogels due to their similar mechanical behavior [167,168]. Additionally, to study the liver, Tsang et al. used PEG hydrogel to model hepatic tissues [169]. PEG can be modified in its mechanical properties and is therefore also suitable for studying the microvasculature [170]. 

#### 3.2.2. Silicon

One of the oldest and most common materials, used since time immemorial in the development of Micro-Electromechanical Systems (MEMS) and later in the microfluidic devices (lab-on-chip), is silicon [171]. It is used in this technology both as a substrate and in the process of microfabrication (sacrificial layers) in various forms as well as single-crystal silicon (SCS), polycrystalline silicon, silicon dioxide, and silicon nitride. It is also widely used in sensors and actuators (especially SCS) [172]. Silicon is well suited for different microfabrication processes, such as etching (wet etching or plasma etching), laser processing, and various bonding methods because it has favorable mechanical properties, and the result is comparatively inexpensive [173]. 

#### 3.2.3. Metals

Another group of materials that is integrated into OOCs consists of metals. As explained in section two, some parameters need to be measured and monitored in OOC experiments. There are some well-designed OOCs that are able to provide real-time data thanks to the sensor systems integrated into their design. Often these sensor systems consist of the metal deposition/integration of metals such as gold and titanium [124]. For example, Henry et al. have developed an OOC with gold electrodes integrated on the substrate (using metal deposition) to measure TEER. The TEER chip has been used for both lung and gut and allows real-time measurements without disrupting the experiment [174]. Titanium is also used to integrate a strain gauge sensor in a heart-on-chip [175].

#### 3.2.4. Membranes

As OOCs recapitulate the cellular interactions in between different compartmental units of the tissue, a boundary is required to separate the cells. This separation is usually achieved by using a porous membrane. This synthetic, permeable, porous membrane helps connect different cells in the tissue and allows material transfer within the compartments of the chips [176]. There are some important properties that need to be considered when designing and fabricating these membranes for integration into OOCs, such as the elasticity (flexibility), transparency, biocompatibility, and cytocompatibility [177]. PDMS, poly(carbonate) (PC), and poly(ethylene terephthalate) (PET) are the most common materials in this regard [178]. Additionally, when designing OOCs, the membrane must be selected in coordination with the substrate materials as there are considerations and protocols for bonding different materials such as PDMS-PET and PDMS-PC [179,180,181]. 

## 4. Fabrication Methods

Microfabrication is the art of miniaturizing devices, which is completely different from conventional machining and manufacturing. Various techniques are used to add or remove materials, pattern the substrate to create the desired geometry, and perform other modification steps. In this section, the common approaches for microfabrication of OOCs are presented. In general, there are two approaches for fabrication of OOCs: (i) bottom-up and (ii) top-down. In the bottom-up methods, the microstructures need to be considered and the cells are seeded into a microenvironment (usually hydrogels) to develop their vascular networks [182]. In the top-down approaches, the microstructure (microvessels) is created and then the cells are seeded. Sometimes a hybrid approach is taken that includes both the bottom-up and the top-down approaches [183]. Below are the techniques used to create the desired microstructures in the device or substrate.

### 4.1. Soft Lithography

Soft lithography is a technique that basically goes back to the well-known method of photolithography, and is applied to a wider range of materials, especially elastomers (such as PDMS) (Figure 6) [184]. The method was offered due to certain limitations of photolithography that were encountered when working with biological systems. The relatively high cost (both for the process and for facilities such as clean rooms), the incompatibility with curved substrates and the lack of surface control are cited as the limitations of photolithography [185]. There is no clear-cut approach to soft lithography, and many methods have been introduced for patterning mechanically soft materials, such as replica molding [185], solvent-assisted micromolding [186], capillary molding [187], phase-shifting edge lithography [188], microcontact printing [189], and micro-transfer molding [190]. Fast prototyping and low cost are the two key advantages of the method, although it is dependent on other lithographic processes such as photolithography to produce the “master” (Master: the main pattern of which is going to be applied on the substrates [191]). 

There are many cases where soft lithography has been employed to fabricate a chip for recapitulating an organ, such as the kidney-on-chip [192], brain-on-chip [193], and gut-and-liver-on-chip [194].

### 4.2. Hot Embossing

It is known as a very suitable and flexible method for the microfabrication of polymer-based chips, such as with the most standard thermoplastic materials such as PMMA [195]. First, a master must be designed and fabricated, which is usually produced by photolithography. Then, in the embossing machine, where the master is mounted, the process is carried out by applying force and heating (either isothermal heating or cooling). The method has many advantages, such as low cost and the possibility of producing a polymeric microstructure with a high aspect ratio and micro-pin lamellae. However, to obtain a high-quality surface, the precise control of the temperature and other influencing parameters is required [196,197].

### 4.3. Injection Molding

Further attempts to the reduce microfabrication processes led to casting and, especially, to injection molding techniques. Injection molding is also applicable for polymers [198]. The first notable advantage is that, as with the photolithography or deposition techniques, the cost of high-precision microfabrication is limited to the fabrication of the master. Many microdevices (e.g., sampling cells, micro-heat exchangers, and some lab-on-a-chip packages) are mass produced using this method. The fabrication is very complex as the temperature, pressure, and injection rate must be controlled to ensure high production quality [199,200]. Lee et al. have used an injected 3D culture scaffold to study angiogenesis by patterning human umbilical endothelial cells (HUVEC) and lung fibroblasts (LF) [201]. They believe the device provides high throughput for vascularized microphysiological environments. 

### 4.4. 3D Printing

The advent of new technologies has created high-resolution, yet low-cost, fabrication capabilities. Three-dimensional material printing is an emerging method for manufacturing 3D templates, components, and devices at the microscale [202]. Interestingly, they are also capable of printing biomedical parts and tissues using cells, matrices, and biomaterials [203]. Three-dimensional printing refers to a set of manufacturing techniques for fabrication, that is, a “layer-by-layer approach” (Layer-by-Layer approach: laying materials from the bottom layers to the top in additive manufacturing [204]). Known techniques include stereolithography, multi-jet modeling, and focused-deposition modeling [205]. It can also be a combination of additive and subtractive manufacturing. In microfabrication, rapid prototyping is a major advantage of this approach as it can replace the master molds made by photolithography and also allows direct fabrication of various other microstructures [206]. In the bioprinting of artificial living organs, there are usually six steps, including imaging, design, material selection, cell selection, bioprinting, and application [207]. The advantages of 3D printing include the precise control and application of the desired cell arrangement. However, there are still limitations in the smallest dimensions that can be achieved (now about a 350 micron channel), and it is also not compatible with all materials [208]. Nonetheless, the use of 3D printing technologies for the fabrication of OOC devices is increasing. For example, Chang et al. modeled a 3D liver by biofabrication for drug testing [209]. They used a PDMS platform fabricated by soft lithography and Hep G2 cells for 3D bioprinting. Additionally, Johnson et al. studied the pseudorabies virus in the nervous system using an organ-level bioprinting platform fabricated by 3D extrusion printing technology [210]. Varone et al. used a 3D printed mold to develop a new design called the open-top spiral chip. The design consists of microchannels and a hydrogel chamber that gives the chip flexibility [211].

## 5. Creative Methods

Apart from the usual microfabrication methods that require high-technology facilities, there are many innovative and low-cost approaches that have been introduced for microfabrication, and many studies have also been carried out with them. The main idea is to make the OOC studies accessible to all laboratories as the OOC is a superior alternative to the usual cell culture systems. However, their fabrication methods today require expensive facilities and clean rooms (Figure 7). Therefore, to compete the available technologies, novel and easy-to-implement methods are being tested that have many positive aspects, such as that of attracting more researchers to the field and reducing costs. To investigate the BBB-on-chip, Sooriyaarachchi proposed a method in which microfluidic devices are fabricated by polycaprolactone-coated sugar microfibers. After dissolving the sugar, the channel remains in the substrate [212]. Additionally, Winkler et al. investigated the feasibility of a tape-based barrier-on-a-chip for small intestine modeling [134]. They presented two adhesive tapes and two fabrication methods with biocompatibility. They concluded that double-sided pressure-sensitive adhesive tapes are both functionally and economically feasible. Moreover, Salman et al. used an acupuncture needle to fabricate an artificial microvessel as a scaffold for a collagen matrix injection [213]. On the other hand, there are some problems that are cited as disadvantages of the approach, such as the large uncertainty in geometry and the limited number of materials that can be used (Table 3) [214].

## 6. Applications of OOCs

Microphysiological systems were invented to create accurate physiological conditions in experiments, which is of the utmost importance for human health and life. Drug development studies play a crucial role in the health-improvement system, and pharmaceutical research is always striving to improve drug efficacy through more accurate preclinical data collected using in vitro and in vivo approaches [215]. The OOCs are potential alternatives to animal models in this chain as they provide highly efficient and economically feasible microphysiological environments for culturing cells from various organisms [216]. Drug delivery, drug screening, disease modeling, and toxicity studies are part of the direct application of OOCs [217,218]. In this context, various organs, ranging from lung to brain, have been modeled and studied using OOCs. The design requirements of each model vary according to the physiological factors and include single or multiple organs, as discussed in the previous sections. Although it is not practical to describe and list all the OOC models in this review, some examples of different chip designs are given when describing a particular organ model.

### 6.1. OOC Technologies

This section presents the available OOC technologies and highlights the important features to consider in their design and fabrication. They go beyond the general specifications that are important for almost all OOC types, such as the transparency of the material and the gas permeability.

#### 6.1.1. Single Organ Models

Brain-on-Chip: The brain is a part of the Central Nervous System (CNS) and is responsible for regulating the body’s activities through signals. It is a rather complex organ that includes many cell types, such as astrocytes, pericytes, and microglia, but most importantly, it is composed of more than 100 billion nerve cells, which are also called neurons [219,220]. Neurons communicate with each other through synapses and also with other cell types to transmit information and signals to organize body functions. Occasionally, there are disruptions in the function of this system, which can occur both naturally (aging) and randomly, by accident. Disorders of the nervous system, categorized as Neurodegenerative Diseases (NDDs), are increasing worldwide, and the World Health Organization (WHO) predicts a mortality rate of 12.22% in 2030 (it was 11.84% in 2015) [221]. NDDs include a number of diseases, such as Alzheimer’s disease (AD), Parkinson’s disease (PD), migraine, and Spinal Muscular Atrophy (SMA) [222]. Unfortunately, there are no effective treatments for many of these diseases [223]. The development of in vitro models is a promising approach to overcome the limited efficacy of available drugs as they provide the opportunity to test therapeutic agents and study disease progression at the cellular and molecular level [224]. In this context, Microphysiological Neural Systems (MPNS) are proposed as one of the promising 3D culture techniques to overcome the limitations of conventional 2D cultures and animal tests [225]. 

Recent studies show that neurons from different parts of the brain do not share the same characteristics, including cell composition, protein, metabolism, and electrical activity [226]. To address this, the multiregional brain-on-chip was introduced as a new approach [227]. They cultured neurons, isolated from the prefrontal cortex, hippocampus, and amygdala of a rat on a single chip and analyzed the cell composition, protein levels, and electrical activities. The results showed good agreement with in vivo tests conducted on rats [228].

Furthermore, various forms of brain disease have been recapitulated using OOC technologies in an in vitro setting. Epilepsy is a CNS disorder related to the electrical activity of neurons [229]. Current antiepileptic drugs are used to suppress or reduce the intensity of the seizures, and there is no clinical therapy [230]. Epilepsy-on-chip systems, in which tissue slices from the hippocampi of Sprague-Dawley rats are placed on metal electrodes, provide a platform for the screening of antiepileptic drugs while measuring neuronal signals [231]. The chip resulted in the finding of promising compounds by providing chronic electrical and optical records. AD is another brain disease for which different approaches are being attempted to understand the main causes. Park et al. developed a 3D brain-on-chip with dynamic conditions and interstitial flows to recapitulate AD conditions and perform more controlled experiments in order to understand the disease mechanisms [232]. A controlled environment to mimic self-organized neuronal differentiation is another issue that is being miniaturized on a chip, and the proposed model by Kilic et al. provides a greater speed in differentiation and improves responses to chemotactic gradients [193]. The ability to image the alive cells under flow conditions and the ability to use electrodes to simultaneously monitor electrophysiological functions are part of the aforementioned benefits of the OOCs and must be achieved during design and fabrication [233].

-BBB-on-Chip: Drugs are not effective in the CNS unless they pass through the highly selective brain microvascular endothelial cells [234]. The blood-brain barrier (BBB) is a combination of Brain Microvascular Endothelial Cells (BMECs) in the capillaries and the surrounding cells in the CNS, which consists of pericytes and astrocytes [235]. Although the BBB blocks numerous drug compounds from entering the brain, it protects the CNS and brain from pathogens [236]. It is expected that BBB-on-chip models based on human cells will be increasingly used in drug-discovery and drug-delivery research on the brain [237,238,239] as the in vivo expression of many solute carriers and efflux transporters varies widely between human and animal systems due to differences between species [59,240]. In the development of BBB-on-chip models for drug screening, the following aspects are crucial: (i) two compartments recapitulating blood and brain parts separated by a porous membrane provide the possibility to sample both brain and blood channels for permeability assays and to directly control and manipulate both the brain and the blood compartments simultaneously [49,50]; (ii) brain endothelial cells mimicking physiological functions, forming a high barrier integrity and expressing efflux pumps, which requires precise control of shear walls (viscosity and flow) to maintain polarity [241]. The literature provides some good examples of BBB-on-chip models that meet both of these criteria [49,50,242]. For example, Park et al. used a sandwiched double channel separated with a porous microfluidic chip to model the BBB within [49]. They used a unique, developmentally inspired iPS differentiation protocol to obtain brain endothelial cells seeded into the bottom channel of the chip to mimic the brain vasculature. Primary astrocytes and pericytes were seeded in the upper channel to mimic brain parenchyma. They demonstrated effective levels of barrier function for up to two weeks and the validation of delivery systems that transport drugs and therapeutic antibodies through the human BBB. Recently, Liang and Yoon used a well-based design of the BBB-on-chip with integrated sensors for sensing TEER, which was shown to be more effective compared to previous designs [242]. -Lung-on-Chip: The interaction between the flow of air during inhalation and exhalation and the blood capillaries of the lungs is an important phenomenon to observe. One tangible reason for its importance is pandemic diseases, such as COVID-19 and influenza, as this is where viral or bacterial infections begin; therefore, physiologically relevant lung models can be used to develop effective drugs and treatments to protect the entire body [243,244]. Transparent, flexible, and low-cost OOCs are one of the best options to perform this type of research and investigate lung issues such as disease etiology and drug screening [40]. In most lung-on-chip designs, there are two channels separated by a porous membrane to recapitulate the microphysiological environment of the lung [76]. 1. Air channel: Lung epithelial cells are cultured in air without flowing media; they are nourished via the adjacent channel. 2. Blood channel: Lung endothelial cells are cultured here under flow conditions to recapitulate blood capillaries. Cyclic respiratory motion is another factor that must be considered when developing a physiologically relevant lung-on-a-chip model to recapitulate breathing motions at an exact rhythm, rate, and magnitude, which has been shown to have drastic effects on tissue function [34]. Vacuum chambers are the solution presented to exert a cyclic suction when combined with an elastic material to mimic this biomechanical motion [36]. Huang et al. presented a new design of a lung-on-chip, integrating gelatin hydrogel into a PDMS structure that can be subjected to cyclic stress to recapitulate the breathing motions [245]. This improves the similarity to the real organ as the mechanical properties as well as the stiffness are close to that of the human lung, and the results also better match the in vivo environment. In addition, Si et al. recently investigated the use of the lung-on-chip to model viral infections and rapidly screen therapeutic candidates [83]. They proposed a human lung bronchial airway modeled on-chip with lung epithelial cells and pulmonary endothelium. The chips were tested with a virus (coronavirus 2 (SARS-CoV-2)), and the best therapeutic was introduced accordingly.-Liver-on-Chip: Drug-induced toxicity is a critical factor in drug development models, and the liver is the organ most vulnerable to potential hazards. Regardless of the research conducted to treat liver disease, studying this organ reduces the number of drug failures. OOCs are considered the best approach for studying the liver because animal studies are expensive, time-consuming, and in some cases inaccurate [246]. Cellular components and biomechanical factors are some of the critical parameters for proper functioning of liver chips. There are several cell types in the liver that maintain the physiological functions, including Kupffer, stellate, and endothelial cells; thus, co-culturing approaches are recommended. Geometry and flow are the most important biomechanical aspects in developing a liver-on-chip. Moreover, liver microvessels are sinusoidal and have mainly rectangular cross-sections [247]. Therefore, when designing vascular sinusoids, aspect ratios and velocities must be accurately calculated to maintain a laminar flow regime (Re <1), which directly affects the compensation of the concentration gradients [43,248]. In this regard, Deng et al. performed a study on liver-on-chips to evaluate hepatoprotective activity [249]. They used a sinusoidal, single microchannel (PDMS-glass) chip seeded with four different hepatic cell lines and perfused with laminar flow. Their observations were promising as they recorded different mechanisms of hepatoprotectants. Kim et al. also used a PDMS-glass chip with straight microchannels and a porous membrane to study liver-on-chips in order to study the metastasis of breast-cancer-derived extracellular vesicles to the liver [246]. -Kidney-on-Chip: The kidney is an organ that balances the body’s fluid and filters the blood. The process of waste removal is an important feature that is closely related to drug composition and toxicity and needs to be monitored accurately [250]. Apart from drugs, there are other conditions that affect the filtration process such as urinary stone disease leading to inflammation, which needs to be thoroughly investigated [251]. Nephrons are small functional units in the kidney that are responsible for purifying the blood [252]. The kidney is composed of various parts, including the glomerulus [253], the proximal [254], and the distal tubule [255], which have been studied individually on a chip. A typical kidney-on-chip has two channels where the urinary lumens are in contact with the interstitial flow. Ultra-filtering is a key consideration in the design of the chip and is tightly controlled by the shear stresses exerted on the cells, which are low (~0.2 dyn/cm^2^) compared to other organs [256]. -Gut-on-Chip: The gut is a multifunctional organ where orally ingested drugs and nutrients are digested, transported, and absorbed. Therefore, it is an important factor in drug efficacy which must be in concert with the barrier function that blocks certain compounds to protect the body [257]. However, the gut is quite a complex physiological environment as other microbial symbionts also work to promote intestinal health [258]. Studying the gut is a step forward in improving the body’s immune function, and OOCs are providing a superior alternative to other approaches such as in vivo animal studies, which have often failed in the transition of the data to the clinic [259]. The relevant literature distinguishes between two types of gut-on-chips: intestine-on-chip [43,260,261] and colon-on-chip [262,263]. The most commonly used gut-on-chip models have typically two channels connected with a porous media; one is intestinal epithelial and the other is vascular endothelial. Accurate barrier function should be achieved for better results in the stage of designing and fabrication. Further cyclic strains and anaerobic environments are sometimes applied in the corresponding research [264].-Heart-on-Chip: Heart disease ranks first among potentially fatal diseases worldwide [265]. For this reason, effective and inexpensive drugs for its treatment are especially important to save the lives of many people. Three-dimensional, bioengineered OOCs of the heart are used effectively for drug testing because they can recapitulate the physiological mechanisms and cell interactions associated with the biomechanical factors [266]. “Cardiac motion” (Cardiac motion: the heart’s cyclic motion with a 0.6–2 Hz frequency as a result of the heart beating (40–120 beats per minute) [267]) is due to highly polarized and contractile cells called cardiomyocytes, and their function is directly related to flow rate, calcium ion concentration, and electrical stimuli [268]. Thus, providing cardiac physiology on a chip requires precise design to perform the mechanical, electrical, and chemical functions [269]. In addition, the design of cardiac chip models must take into account the ability of the chip to perform contractility techniques such as muscular thin films and to acquire electrophysiological and morphological data [270]. For example, Liu et al. used a double-channel microfluidic device made of PDMS to model a human heart-on-chip [86]. Their heart-on-chip was lined with vein endothelial cells, induced pluripotent stem cells, and fibroblasts (gingival fibroblasts). The model is expected to be a functional tool for pharmacological studies and personalized medicine.-Bone-on-Chip: Bones are living tissues that both serve as the structure of the body and produce the major blood cells [271]. It has three main tissues (compact, cancellous, and subchondral), in which different types of bone cells (osteoblasts, osteoclasts, osteocytes, and hematopoietic cells) maintain bone metabolism and blood cell production [272]. Cancellous bone tissue consists of a spongy substance called *marrow*, which is responsible for blood production in the middle of the bone. Chou et al. have introduced an in vitro model of “bone marrow” (Bone marrow: a sponge-like tissue inside the bone which produces diverse materials, including stem and blood cells [273]) using microfluidics to study toxicities and dysfunction caused by factors such as drugs and radiation. Their chip consists of two channels representing the vasculature and hematopoietic system separated by a porous membrane. They obtained promising results for studying responses to drugs and also to radiation [274]. In another study, Bahmaee et al. presented a new study consisting of a microfluidic device (bioreactor) and a scaffold chamber with a hexagonal pillar pattern to study osteogenesis-on-chip [275]. They claimed that their device is a new and effective platform for testing bone drugs compared to the usual approaches in this field. Additionally, there are related research trends using bone-on-chips to study bone metastasis and metastasis colonization for the purpose of cancer treatment and prevention [276].-Other Organs: The developing OOC models include different parts and bring revolutionary breakthroughs compared to the previous trend. The skin is the first external organ that protects the body and is very likely to be affected by chemical substances, pollutants, and Ultraviolet Light (UV); thus, conducting research to protect, prevent, or cure corresponding diseases is very important. Previously, optically visible skin layers were studied on chips to mimic the interactions between layers and to investigate the biology behind them [277]. Wufuer et al. designed a three-layer chip representing epidermal, dermal, and endothelial cells to recapitulate the dense skin barrier [278]. They were able to study the drugs and concluded that the chips were suitable for modeling inflammatory skin diseases. In recent studies, skin-on-chip modelers have been looking for new approaches to add hair follicles, sweat glands, and pigmentation for more advanced research [279]. 

OOC technology has been used to create numerous organ models, and the number of these models is increasing daily [280,281]. The vagina-on-chip (VOG) [282], to model the female reproductive system, and the eye-on-chip (EOC) [283] for drug trials in eye diseases are some of the most interesting new OOC models [284]. VOGs investigate changes in cells due to metabolic activities and also make observations of hormonal and chemical treatments on fertility [285]. Different types of chips have been introduced in this regard, including the placenta-on-chip [286], the uterus-on-chip [287], and the endometrium-on-chip [288]. Moreover, EOCs have successfully tested for mimicking Choroidal neovascularization (CNV) which is counted as an important threat and a reason for visual loss [284]. A recent breakthrough in this field is the cornea-on-chip with a blinking eye for an accurate evaluation of the drugs [289]. The immune-on-chip is also a new emerging topic in this field, on which Polini et al. presented a comprehensive review [290]. The design of a lymph node chip requires several considerations, including the provision of a biomimetic scaffold on which the immune cells can be located, a microenvironment with the appropriate ECM components, and dynamic flow [291]. Furthermore, lymph-node-on-chips have been studied by Shanti et al., who describe their design and implications for drug discovery and delivery [292].

#### 6.1.2. Multi Organ Models

Given the extensive interactions between organs in the body, an independent organ-on-a-chip cannot fully recapitulate the entire complexity of an organism [293]. In recent years, an unprecedented attempt has been made to integrate OOC technology for other organs through the human-on-chip projects (Figure 8) [294]. Several attempts were made to recapitulate this complexity by connecting multiple organ chip models via their vascular channels as in the in vivo systems. In drug development studies in particular, it is especially important to examine how the body handles the drug; thus, the duration and intensity of the drug’s effect must be also monitored. This is known as pharmacokinetics (PK) and is accompanied by pharmacodynamics (PD) to capture the biochemical and physiological aspects of the drugs [295]. As human PK and PD values are difficult to predict with other in vivo systems based on different organisms, linked microfluidic devices are proposed as an effective alternative [296]. Novak et al. succeeded in the chronic monitoring of eight vascularized, double-channeled OOCs comprising heart, liver, kidney, intestine, skin, BBB, lung, and brain. It was facilitated by a liquid-handling robot to achieve an automated culture system [297]. Choe et al. fabricated a connected gut and liver chip model (gut-liver-on-a-chip) and studied the first-pass metabolism to improve drug efficacy [194]. Another achievement in this field, made by Schimek et al., is that of the co-culturing of lung and liver cells on the same chip environment, which was studied for access to inhaled toxic substances [298]. Skardal et al. studied three different tissues on a chip, including the lung, liver, and heart to monitor drug response [53]. Gradually, major organs and other tissues are being incorporated into this approach. Increased accuracy of the results and predictions of drug efficacy, as well as cost and time savings, are reported as the main incentives for developing multi-organ-on-chip technologies [299].

## 7. Discussion

Organ-on-chips provide a novel and unique platform for research on various diseases by contributing to both diagnosis and treatment approaches, which are important research areas for improving global health. The time and cost savings in drug development is a strong motivation for this technology, as is the elimination of in vivo animal testing. There are specific features that make it a potential alternative and breakthrough tool in the field. These include 3D culture and cell–cell interactions, the ability to apply mechanobiological stimuli, online monitoring of testing, and low testing cost.

On the other hand, critics of OOCs have pointed out how accurately the technology can recapitulate microphysiological environments and ask whether the existing uncertainties affect the final results. In addition, concerns were raised that the examination of a single organ represented by a chip is not crucial for drug testing, even if the outcome is effective [300]. Debates have been held on the effectiveness of OOCs, and many practical solutions have been proposed to overcome the shortcomings, but development is still ongoing. To overcome the challenges and also to mention the promising features of OOCs, a thorough evaluation is needed from the first step of the design process to the microfabrication of the chips, as shown in Figure 9.

The design of a chip is a combination of physiological and mechanical concepts governed by material and fabrication concepts. An organ-on-chip cannot fully recapitulate all the features of an organ, but only the most important functions that are crucial in terms of the physiological concepts and based on the scientific question being asked. Thus, the design is closely related to the goal of the research and simplifications are required to develop a viable design.

A fundamental issue that needs to be addressed is whether a simple microchannel can meet the requirements of an OOC. To answer this question, it is important to mention that microchannels are scaffold-like structures that are best suited for the study of vessels when accompanied by appropriate cell culture procedures. Flow is an important element of an OOC. Flow maintains several biomechanical factors, including shear stress and interstitial flow, while regularly providing nutrients and oxygen to cells and stimulating the organ to live longer and maintain its properties for testing. In addition, layered structures provide the opportunity to study permeability, which is important in the field of drug delivery. Porous membranes are the common element to help in recapitulating this property of organs The use of flexible materials also makes it possible to mimic the mechanical movements of organs, such as during breathing. Although a microchannel can help to mimic an organ and bring with it the requirement to culture and stimulate it, it cannot function on its own without suitable packaging for monitoring.

The physiochemical properties of the flow, such as pH and dissolved oxygen, can change during testing and have a large effect on the biological performance, so that deviations exhibit erratic behavior. The use of online monitoring techniques is a prerequisite for improving the accuracy. Conventional or innovative microelectromechanical on-chip systems can be selected or designed for packaging [301].

The materials are another crucial factor affecting the whole process of design and fabrication. The materials should not interfere with the experiments; so, they must be compatible with the culture environment and test compositions. The selection of materials according to the desired criteria make microfabrication more complex and also more expensive. Therefore, a number of common materials which have some basic properties, such as gas permeability, optical clarity, and rapid prototyping, are usually of interest [302]. One possible solution in this regard is surface modification approaches that alter the material to have little impact on the test compositions, even if some of them are temporary and do not cover the entire time of the experiments.

A standout piece in this flowchart is microfabrication. It infringes the liberty of OOC research by prohibitively expensive facilities. Soft lithography is a widely used, fast, and economically feasible technique that also requires one step photolithography and clean room facilities. However, as mentioned earlier, it only provides the microchannel, and other techniques are required for on-chip measurement devices, which make it expensive. Furthermore, it has been shown that for bonding the substrates, again certain materials and techniques work perfectly [303], which proves how much the microfabrication techniques and materials influence the OOC design. The utilization of sophisticated fabrication methods comes at the price of low throughput and the expensiveness of the chips. Therefore, in parallel to sophisticated fabrication methods for OOCs, simple approaches have been proposed to develop the OOC in less equipped laboratories that have shown acceptable results [134]. However, in reality these approaches have even more limited materials to use, along with increased geometrical uncertainties resulting from simple fabrication. It can be stated that they can be used for preliminary studies, but currently advanced chip designs are inevitable for more accurate results. Both academic laboratories and startup companies have made significant progress in mass-producing chips. Although a single commercial chip can currently cost as much as several conventional 2D/3D culture systems, new and cost-effective chip designs are being introduced onto the market every day.

The aforementioned benefits suggest that OOC technology is a promising tool to rely on and that ongoing interdisciplinary research will lead it to play a key role in the future. Although the results are already considerable, there are limited opportunities to design and fabricate different chips, which shows the gap for finding compatible materials and microfabrication techniques. In addition, online monitoring and on-chip measurement devices are another area to move into for more reliable results. To further study the dynamics of the vasculature, as research is being conducted in this regard, such as ultrasound exposure, an ordinary PDMS-based OOC has defined microchannels surrounded by materials, so that the walls are fixed, and it is not possible to study the dynamic motion on them. Other techniques and materials are needed, such as hydrogels-based chips or a combination thereof. Moreover, other mechanical stimuli may be of interest, including temperature and pressure, which are always considered in flow control.

## 8. Conclusions

Organ-on-chips are expected to become an influential technology by providing physiologically relevant mechanobiological environments suitable for disease research and drug development. A significant advantage of OOCs is the control over geometry that supports the vascular study in OOCs, which was nearly impossible with the previous technologies. Moreover, the control of flow properties along with biomechanical factors improves in vitro test results and makes it a preferred approach [63]. The chips have shown promising results, although they are a new approach that comes with a variety of challenges. The concerns expressed relate to the reliability of the chips, which needs to be improved through better design and fabrication techniques. There is always a trade-off between the concepts involved in the design process as well as the biological aspects and materials; so, these cannot be fully addressed due to the limitations imposed by microfabrication. Hence, there is a great need for the development of interdisciplinary investigations to expand the possibilities for design while reducing the costs. Data monitoring is another issue in this context that requires the development of innovative on-chip devices to control the input and output and consequently to improve the reliability of the results. Moreover, different designs and the use of new materials could add further stimuli to establish simplification hypotheses that will be used to develop a viable chip.

## Figures and Tables

**Figure 1 micromachines-12-01443-f001:**
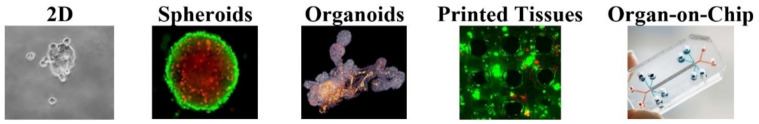
In vitro studying technologies.

**Figure 2 micromachines-12-01443-f002:**
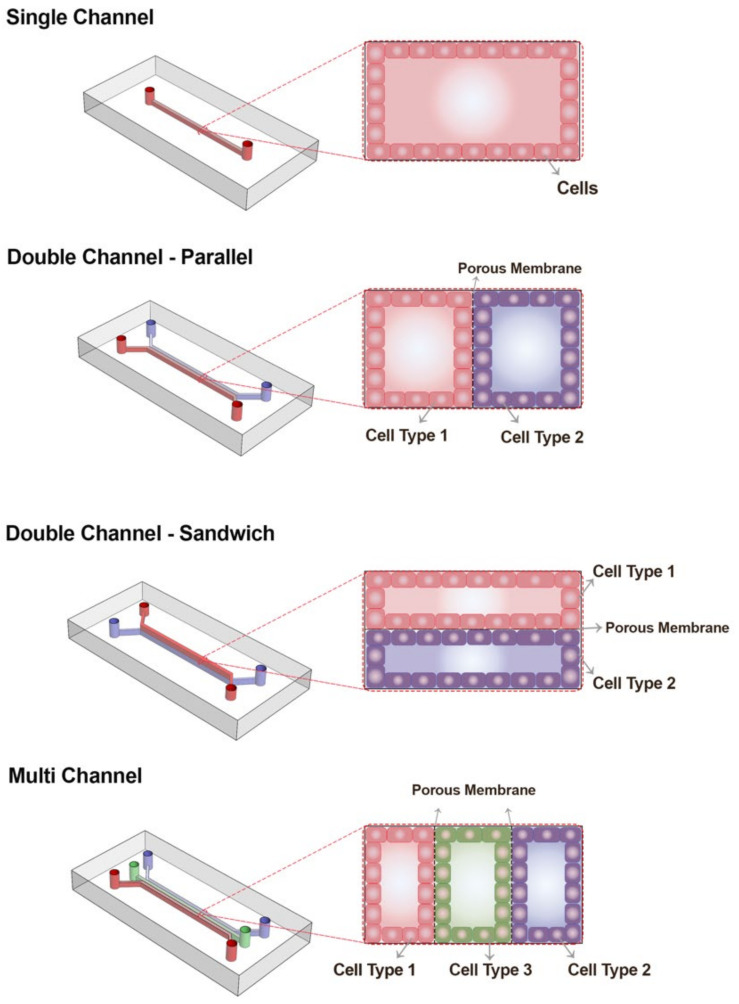
Schematic of an organ-on-chip with one, two, and multi channels. Usually there are three different layers of an OOC: bottom channel, porous membrane, and top channel.

**Figure 3 micromachines-12-01443-f003:**
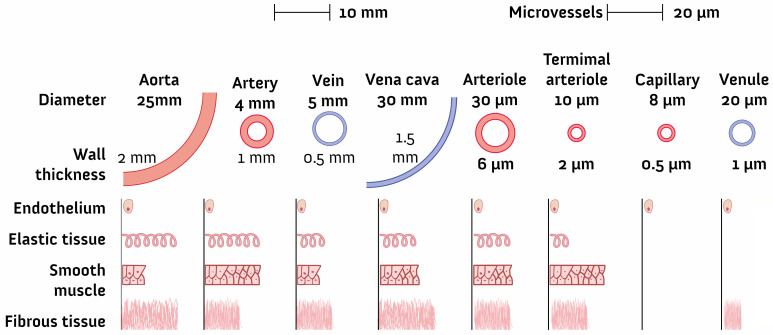
Blood vessel dimensions [73]; approaching real-size OOCs with dimensions less than 20 µm is demanding and impossible without considering other properties as well as flexibility.

**Figure 4 micromachines-12-01443-f004:**
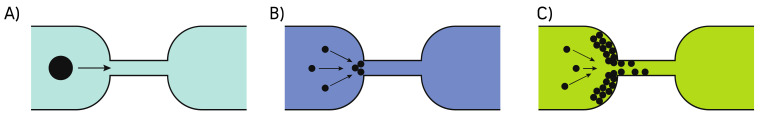
Clogging mechanisms: (**A**) sieving, (**B**) bridging, and (**C**) aggregation.

**Figure 5 micromachines-12-01443-f005:**
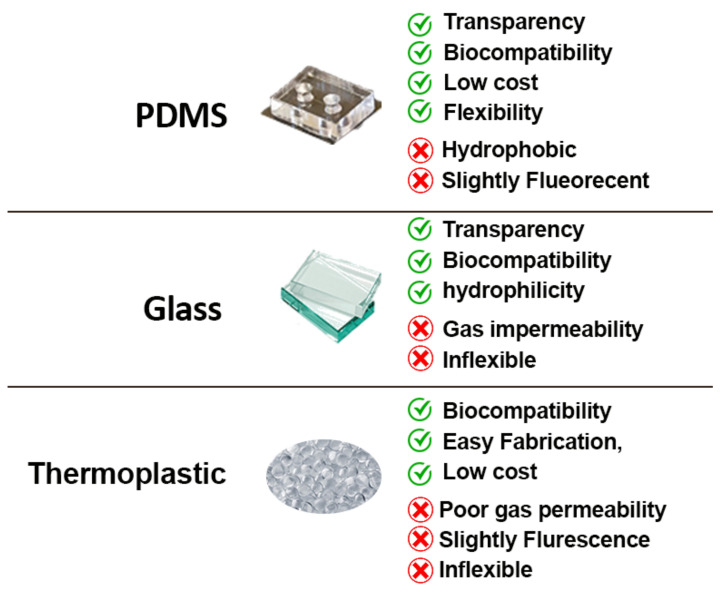
Common materials used for OOCs. The most important features for selecting the materials for the structure are biocompatibility, transparency, and the cost.

**Figure 6 micromachines-12-01443-f006:**
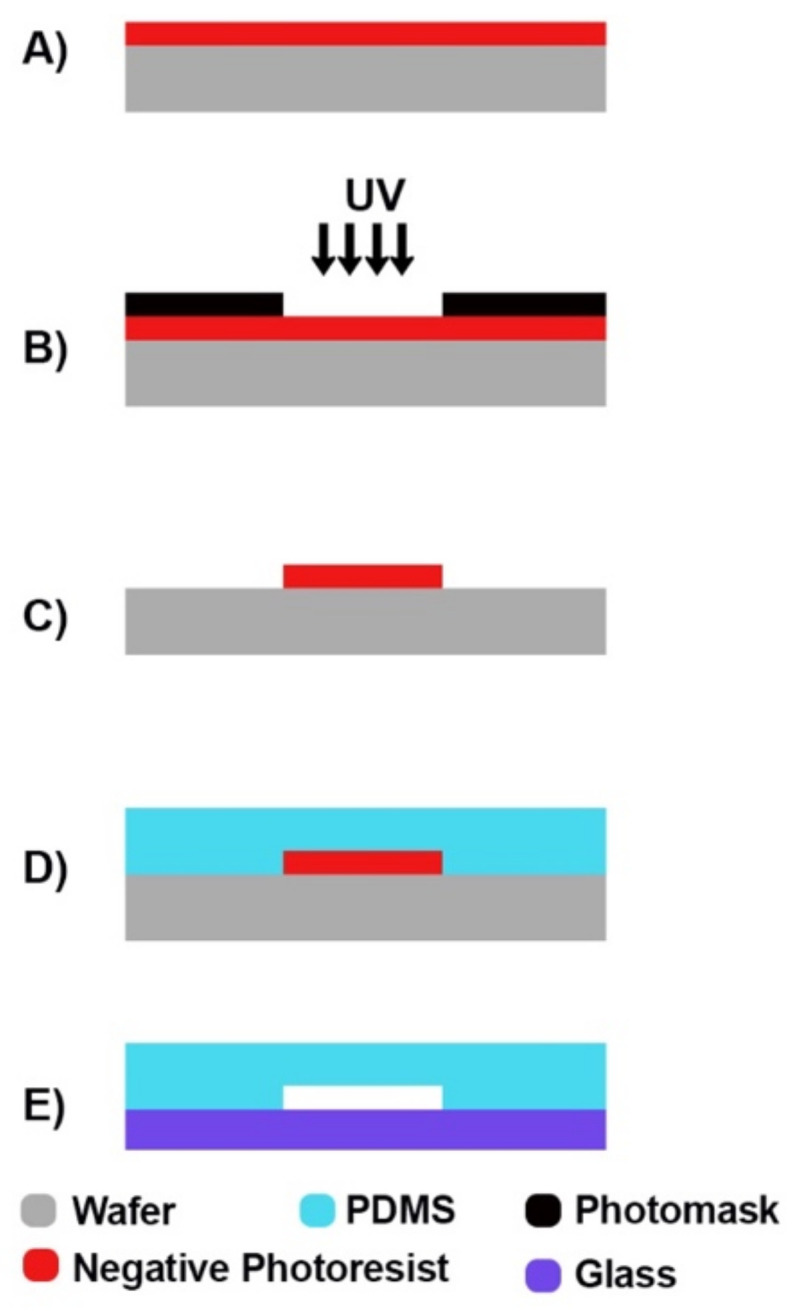
Soft lithography steps for “*negative photoresist*”: (**A**) photoresist coating, (**B**) exposure to UV light through photomask, (**C**) applying developer, (**D**) PDMS molding, and (**E**) bonding.

**Figure 7 micromachines-12-01443-f007:**
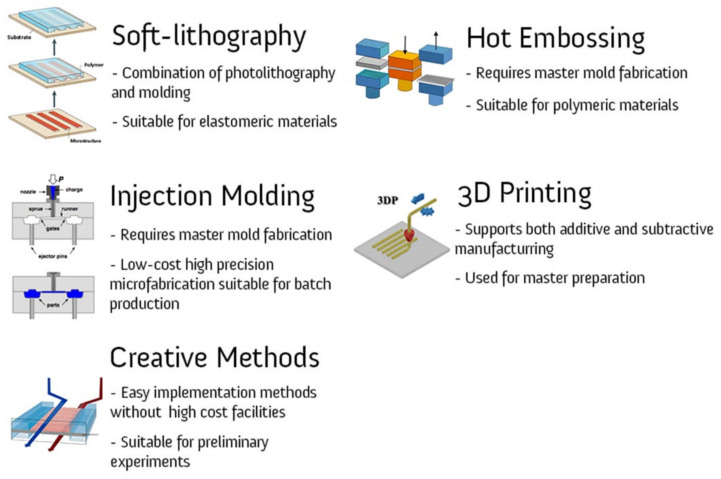
OOC microfabrication methods, including rapid prototyping (3D printing) and batch production (injection molding) methods.

**Figure 8 micromachines-12-01443-f008:**
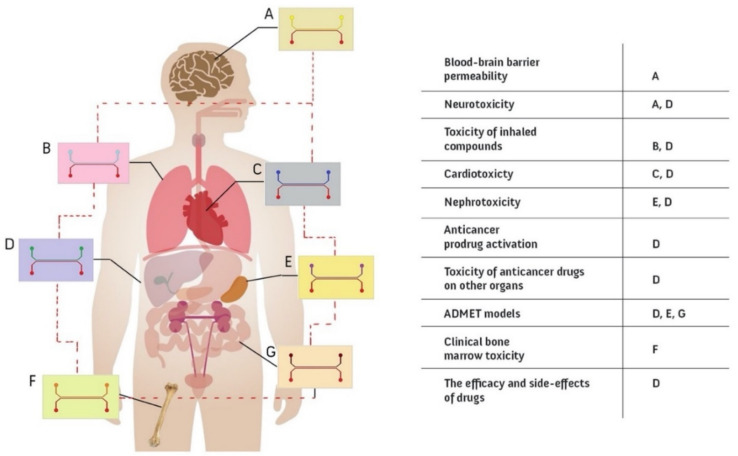
Organ(s)-on-chip; chips recapitulating diverse organs applicable for a wide range of research, including drug efficiency, disease, and ADMET (Adsorption, Distribution, Metabolism, Excretion, and Toxicity) modeling.

**Figure 9 micromachines-12-01443-f009:**
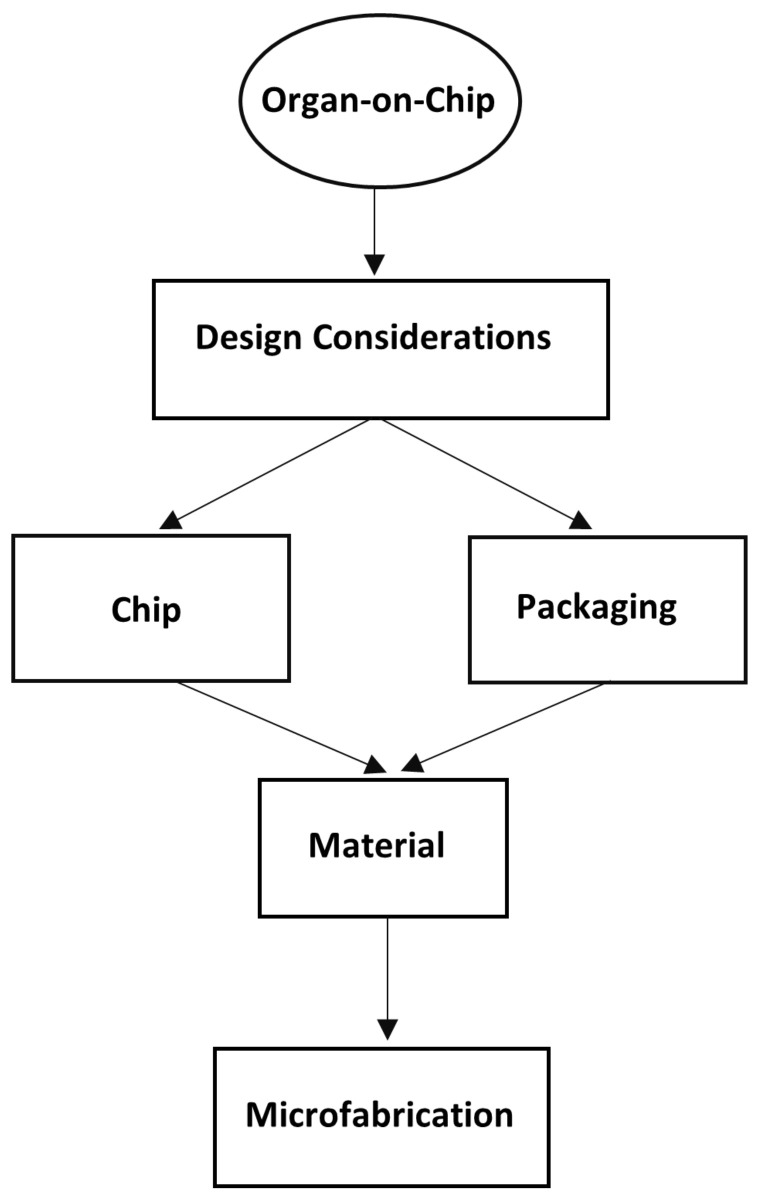
Design and fabrication flowchart of OOCs.

**Table 1 micromachines-12-01443-t001:** Applied flow rates in different OOCs.

Studied Organ	Flow Rate	Refs.
Blood-Brain Barrier	100 µL/h	[49]
	10 µL/min	[78]
	16 µL/min	[79]
	2.5 mL/h	[80]
Lung	30 µL/h	[81]
	60 µL/h	[82,83]
Gut	30 µL/h	[84,85]
	60 µL/h	[44]
Heart	40 µL/h	[86]

**Table 2 micromachines-12-01443-t002:** Different types of micropumps.

Ref.	Type	Available Flow Rate Range (mL/min)
[89]	Peristaltic micropump	1.66 × 10^−4^–3600
[90]	Syringe pumps	1 × 10^−6^–0.127
[91]	Electrokinetic pump	1.8 × 10^−3^–0.01
[92]	Capillary pump	5.05 × 10^−4^–210

**Table 3 micromachines-12-01443-t003:** Comparison of different microfabrication techniques [116,117].

Method		Cost	Facility requirement	Precision	Capability for surface treatment
3D printing	Additive, Removal, and Patterning				
Soft lithography	One step patterning and removal and molding				
Hot Embossing	One step patterning and removal and molding				
Injection molding	One step patterning and removal and molding				
Miscellaneous	---				


 Positive; 

 Moderate; 

 Negative.

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
