# Peer review of "Design and Fabrication of Organ-on-Chips: Promises and Challenges"

_micromachines, 2021, doi:10.3390/mi12121443_

Round 1
Reviewer 1 Report
Attachment

Reviewer 2 Report
This review article provides a good overview of different organ-on-chip systems, design/fabrication strategies and materials used in manufacturing of OOC platforms. The review provides timely information on what has been done in this area as well as the challenges in the current design and manufacturing approaches. The manuscript has been drafted in an organized fashion and it is easy to read and comprehend, with minimal typographical errors throughout. The reviewer suggests the following modifications:
- Given the importance of immune system/cells in regulating the function of almost all organs, it would be great if authors mention the recent advances in immune-on-chip systems (such as Lymph node-on-Chip) or at least the role of immune cells in other OOCs.
- There are still challenges in scaling up, throughput and fabrication costs of OOC designs. Would be beneficial if authors suggest new strategies to address those in the near future.
- Minor comment: line 674; 5-6. “3D” printing
Round 2
Reviewer 1 Report
The comments have been addressed properly.